# Effects of Prolonged Pomace Contact on Color and Mouthfeel Characteristics in Merlot Wine During the Ageing Process Under Microwave Irradiation

**DOI:** 10.3390/foods14030507

**Published:** 2025-02-05

**Authors:** Jiang-Feng Yuan, Hui-Min Qin, Li-Juan Wang, Xiao-Wen Yang, Yang Li, Ning-Bo Wan, Jie Zhang

**Affiliations:** 1College of Food and Bioengineering, Henan University of Science and Technology, Luoyang 471023, China; huiminqin914@163.com (H.-M.Q.); xiaowenyang8@163.com (X.-W.Y.); liyang@haust.edu.cn (Y.L.); ningbo_wan@163.com (N.-B.W.); zhangjie@haust.edu.cn (J.Z.); 2College of Basic Medical Science, Ningxia Medical University, Yinchuan 750004, China; mnn717@163.com

**Keywords:** microwave irradiation, Merlot wine, prolonged pomace contact, color characteristic, mouthfeel characteristic

## Abstract

Wine color and mouthfeel are essential organoleptic characteristics considered by consumers. In this paper, the potential impacts on color and mouthfeel characteristics in wine, without pomace or prolonged pomace contact after different microwave treatment times, were investigated during storage. The results indicated that the trend changes in color and mouthfeel related parameters (including visible spectrum, brightness, red hue, yellow hue, color difference, saturation, hue angle, total polyphenol content, total monomer anthocyanins, total tannins, total flavan-3-ols, epigallocatechin, catechin, epicatechin, epicatechin gallate, and fluorescence spectrum) after microwave-treated and natural aged wines without pomace and prolong pomace contact were very similar. Moreover, changes in these organoleptic parameters of microwave-treated wine were faster than those of untreated wine, which required a long aging time in traditional processing. Also, microwave treatment had a long-term influence on color and mouthfeel characteristics. All these results showed that prolonged pomace contact technology could obviously improve the clarity and yellowness of wine, and microwave technology could reduce wine aging and rapidly change its color and mouthfeel characteristics. In summary, prolonged pomace contact technology is an interesting strategy to replace traditional fining agents. Microwave technology, as an efficient artificial aging technology, might reduce aging time and rapidly change organoleptic characteristics for producing high quality wine.

## 1. Introduction

Wine is a versatile and easily accessible alcoholic beverage that is enjoyed by consumers around the world. There are many varieties of wine, among which Merlot is the most widely planted predominant variety, covering over 62% of the wine market [1]. The quality of Merlot wine is a concern of wineries. To meet consumers’ demands for quality and taste, wineries need to produce high-quality wines, and the wine’s quality depends on physicochemical characteristics, such as alcohol concentration, residual sugar concentration, categories and concentrations of phenolic compounds, concentrations of higher alcohols, aromatic compounds, etc. The hundreds of compounds present in wine relate to its quality, thus determining its sensory characteristics [2]. These include phenolic compounds which are critical for wine quality, determining the wine’s sensory characteristics such as appearance, color, bitterness, astringency, flavor, and stability, ranging from simple phenolic acids to high molecular weight polymers such as condensed and hydrolysable tannins [3]. Phenolic compounds present diverse structures in wine and react with other compounds to form more stable pigments, leading to wine color changes through reducing the content of phenolic compounds and increasing the content of pigments. In contrast, the aggregation of polyphenol compounds could cause a loss of aroma compounds through intermolecular interactions [4]. Flavonoids are the main phenolic compounds that exist in the extensive family of phenolic compounds in wine, including anthocyanidins, flavan-3-ols, and tannins, which are the most important for wine quality due to their contribution to wine sensory characteristics [5].

Pomace is one of the most valuable and abundant by-products in wineries, containing a large amount of phenolic compounds. The technology of prolonged pomace contact has received attention during winemaking operations [6]. One role of prolonged pomace contact is to prolong the extraction time of phenolic compounds and change the medium type, because some phenolic compounds, such as protocatechuic acid or gallic acid, require a longer maceration time to achieve higher extraction concentration. So, prolonged pomace maceration time is used to raise the extraction rates of less water-soluble compounds. The other role of prolonged pomace contact is to enhance wine clarity and stability through the use of pomace as a fining agent for the visible spectra [6]. Traditional fining agents (including chemical fining agents, animal proteins agents, and plant proteins agents) are used to remove unwanted wine components (including phenolic compounds and tannins). This removal causes the loss of color, astringency, and bitterness in wine [2]. Furthermore, adding chemical fining agents and plant protein agents could result in economic waste in the winemaking industry, and the addition of animal protein agents could trigger allergies or intolerant reactions [7]. For these reasons, grape pomace as a fining agent was investigated by winemakers and wine scientists. Compared with commercial fining agents, pomace could efficiently improve wine clarity. Therefore, the use of pomace as a new alternative fining agent for unavoidable waste is actively encouraged [8].

The differences in fresh wine and high-quality wine are due to changes in sensory properties such as color, mouthfeel, stability, clarity, and aroma. Traditional winemaking technology matures wine in oak barrels for a few years, or even longer, for oxidation reactions to form aged wine for consumers. As such, the traditional winemaking process leads to high cost, high oak barrel usage, potential microbial pollution, and a long production cycle. To produce high-quality wine in a short aging time, some artificial physical technologies (such as micro-oxygenation, magnetic field, high hydrostatic pressure, ultrasonic irradiation, microwave irradiation, etc.) have been proposed and studied [9,10]; however, these technologies are still at the pilot-scale stage, and further research is needed to fully understand their industrial-scale application. Among these technologies, microwave technology is an emerging in the winemaking industry due to its relatively low cost, high efficiency, shorter processing time, and pasteurization effect [11], which overcomes the shortcomings of traditional winemaking technology. On the one hand, the thermal effect of microwaves supports the process of accelerating chemical reactions. This principle was based on the influence of polar materials and solvents governed by ionic conduction and dipole rotation, and the thermal effect could be explained by the phenomenon of the increasing extraction rate of total phenolics in the wine maceration phase [12]. On the other hand, the non-thermal effects of microwaves could not be explained by the temperature change, such as the reduction in phenolic acids [13], reducing the SO_2_ addition, the change in PPO on activities and properties [14], or the enhancement of overall wine quality under microwave irradiation [15]. The mechanisms of these phenomena are being explored. An increase in free radicals induced by the breaking bonds after using the microwave seemed to explain the non-thermal effect [16]. The mechanism and results of the non-thermal effect were worthy of further study for potential industrial-scale application. As an emerging technology in wine industry, microwave irradiation could efficiently produce high-quality wine compared with traditional aging process, which could shorten the aging time, reduce production costs [11], reduce the endogenous yeast population [17], reduce the higher alcohol content, change the mouthfeel [18], and improve the overall wine quality by changing the color characteristics [11,19].

Prolonged pomace contact is an interesting strategy for modifying the sensory characteristics of wine by changing its chemical composition [6], and microwave aging technology can rapidly change the sensory characteristics of wine by accelerating the oxidation and polymerization reactions between compounds. To our knowledge, there have been no reports on the dynamic changes in color and mouthfeel characteristics in Merlot wine under microwave irradiation combined with the technology of prolonged pomace contact during the aging process. In this study, the color characteristics, total polyphenol content, total monomer anthocyanins, total tannins, total flavan-3-ols, main flavane-3-ol compounds, fluorescence properties, and sensory evaluation were analyzed after microwave treatment combined with the technology of prolonged pomace contact during fresh Merlot wine storage. The study aimed to prove the positive effects of microwaves on modifying color and mouthfeel characteristics in wine, analyze the outcomes of prolonged pomace contact on the fining process, and investigate the effect of microwave technology combined with prolonged pomace contact on the sensory properties of Merlot wine. All these studies can contribute to understanding the changes in Merlot wine sensory properties caused by microwave aging technology and prolonged pomace contact technology.

## 2. Materials and Methods

### 2.1. Reagents

Gallic acid, epigallocatechin (EGC), catechin (C), epicatechin (EC), epicatechin gallate (ECG), and bovine serum albumin were purchased from Hefei Bomei Biotechnology Co., Ltd. (Hefei, China). A Folin–Ciocalteu reagent, K_2_S_2_O_5_, and pectinase were purchased from Lanji Technology Development Co., Ltd. (Shanghai, China). The p-DMACA reagent was purchased from Jieshikai Biotechnology Co., Ltd. (Shanghai, China). Hydrochloric acid, tartaric acid, sodium carbonate, sodium hydroxide, ethanol, hydrochloric acid, sodium acetate, potassium chloride, glycerol, methanol, acetic acid, and potassium hydrogen tartrate were purchased from Zhiyuan Chemical Reagent Co., Ltd. (Tianjin, China). Acetonitrile (HPLC-grade) was purchased from Thermo Fisher Scientific (Madison, WI, USA). Other reagents and solvents were analytically pure.

### 2.2. Winemaking Process and Experiment Condition

About 50 kg of Merlot grapes at full maturity (5.85 g/L titratable acid in terms of tartaric acid, pH 3.42) were harvested in 2022 from Huailai Vineyard and transported to the Experimental Winery of School of Food and Biotechnology (Henan University of Science and Technology, Luoyang, China). Grapes free from stems were washed and crushed, sulfated (80 mg K_2_S_2_O_5_/kg), then the mixed solution of the musts (skins, seeds, flesh, and juice) with sucrose (1:10, g/kg) was stored in 10 L lab fermentation glass jar. According to the manufacturer’s instructions, we inoculated 200 mg/L commercial saccharomyces cerevisiae (Shanghai Anqi yeast Co., Ltd., Shanghai, China) at 25 ± 2 °C for alcohol fermentation. We pressed the pomace down twice a day (1 min in the morning and 1 min in the afternoon) for cap management. After about 15 days, a spontaneous malolactic fermentation was followed at 18 ± 2 °C. The closed magnetic stirrer and controlled temperature cooling microwave irradiation system (CMCC-MI system) was used to treat Merlot wine. The CMCC-MI system included a microwave reactor (XH-MC-1 microwave synthesis, Xiang Hu Science and Technology Development Co., Ltd., Beijing, China) and a low-temperature coolant circulating pump (Gongyi Yuhua Instrument Co., Ltd., Gongyi, China) [20]. The experimental design consisted of eight treatments (two factors) in triplicate (Figure 1). Two aging strategies (Factor 1, without and with prolonged pomace contact) combined with microwave conditions (Factor 2, different microwave time at 500 W and 40 °C) were applied. The wines without pomace were carried out after alcohol fermentation through filtering skins and seeds. The fresh wine was stored in fermentation glass jars, and then treated daily (continuously for 15 d) with different microwave conditions. The wines with prolonged pomace contact were directly treated daily (continuously for 15 d) with different microwave conditions. After microwave treatment, on the 1st, 20th, 40th, 60th, 80th, 100th, and 120th storage days, the parameters of color and mouthfeel were determined by taking a certain volume of wine samples.

### 2.3. Spectrophotometric Parameters

According to the literature, with some modification [11], all wines samples were diluted 5-fold with 12% (*v*/*v*) ethanol and placed in a 1.0 cm quartz colorimetric cell. Then, they were scanned by a UV-2600 Ultraviolet–Visible spectrophotometer (Tsudo, Japan) in the wavelength range of 380 nm to 800 nm, measured the absorbance at 420, 520, and 620 nm, respectively. The total color intensity (CI) was the sum of absorbance at 420, 520, and 620 nm.

### 2.4. CIELab Parameters

The CIELab space was assessed by SC-80C automatic colorimeter (Beijing Kangguang Instrument Co., Ltd., Beijing, China). The brightness (*L**), red hue (*a**), and yellow hue (*b**) were determined. The color difference (Δ*E*ab*), saturation (*C*ab*), and hue angle (*hab*) were calculated based on the CIELab formulae [11]:$$∆E^{*}=\sqrt{\left(∆L^{*2}+∆a^{*2}+∆b^{*2}\right),}\;C^{*}ab=\sqrt{\left(a^{*2}+b^{*2}\right),}\;hab={tan}^{-1}\left(\frac{b^{*}}{a^{*}}\right).$$

### 2.5. Analysis of Total Polyphenol Content (TPC)

Folin–Ciocalteu colorimetry was used to determine TPC according to the literature [19] with some minor modifications. We added a 25 µL wine sample, 975 µL deionized water, and 0.1 mL Folin–Ciocalteu reagent, then added 3 mL of 5% sodium carbonate solution to the mixture. We incubated the obtained mixture at room temperature in the dark for 60 min, then the absorbance was measured at 765 nm. The result was represented with the equivalent of gallic acid per liter of red wine.

### 2.6. Analysis of Total Monomer Anthocyanins (TMA)

TMA was determined by using pH differential indication according to the literature [19] with some minor modifications. As a test sample, 1.0 mL red wine was diluted with 4.0 mL of 12% ethanol solution. We took 1.0 mL of the test sample, and added 3 mL KCL-HCL buffer solution (pH 1.0) and 3 mL NaAc-HAc buffer solutio (pH 4.5), respectively. We placed it in the dark for 100 min at room temperature. The absorbance was measured at 520 nm and 700 nm, respectively. The results were expressed in mg/L of cyanidin-3-glucoside (cyd-3-glu) equivalents, as follows:$$C=\left(\frac{A\times MW\times DF\times 1000}{\varepsilon \times 1}\right),$$
where A = (A_520nm/pH1.0_−A_700nm/pH1.0_)−(A_520nm/pH4.5_−A_700nm/pH4.5_); MW (molecular weight) = 449.2 g/mol of cyd-3-glu; DF = dilution factor; l = pathlength in cm; ε = 26,900 molar extinction coefficient; and 1000 = conversion from g to mg.

### 2.7. Analysis of Total Tannins (TT)

According to the literature [21], the acid-splitting method was used to determine TT. We added 3 mL of 12 mol/L HCl to 1 mL of diluted wine (1:50) into a tightly sealed glass test tube in the dark. It was heated with a water bath for 30 min, then rapidly cooled. We added 0.5 mL of ethanol and homogenized it. The absorbance was measured at 550 nm (A_550_). Except for removing the heating step, all other process steps were the same, and the absorbance was measured at 550 nm (A′_550_). TT was expressed as: TT = (A_550_−A′_550_) × 19.33.

### 2.8. Analysis of Total Flavan_-3-ols_ (TF_-3-ols_)

TF_3-ols_ was determined by using the p-DMACA method according to the literature [22]. A diluted wine sample of 1.0 mL (1:20) was placed in a glass test tube. We added 0.15 mL glycerol and 5.0 mL p-DMACA, then methanol was added to increase the volume to 10 mL. After 7 min, the absorbance was measured at 640 nm against blank methanol. The p-DMACA reagent was prepared before use, containing 1% (*w*/*v*) p-DMACA and hydrochloric acid solution (4/1, *v*/*v*) in a cold mixture of methanol. Catechin was used as the standard and TF-3-ols was expressed as catechin equivalents (mg/L).

### 2.9. Analysis of Main Flavane-3-ol Compounds

HPLC analysis was carried out using a ZORBAX SB-C18 column (5 μm, 4.6 mm × 250 mm, Agilent, Santa Clara, CA, USA), with the Agilent 1260 infinity HPLC system. After mobile A and mobile B were filtered through 0.45 μm membranes, they were ultrasonically degassed for 20 min. All wine samples were filtered through 0.22 μm filtering membranes. The HPLC operation parameters were as follows: flow rate, 1.0 mL/min; column temperature, 30 °C; injection volume, 10 μL. The H_2_O containing 1% acetic acid was mobile A and acetonitrile was mobile B. The elution gradient program was as follows: 0–10 min, 10% B; 10–15 min, 10–20% B; 15–26 min, 20–40% B; 26–30 min, 40–100% B; 30–35 min, 100–10% B. The flavane-3-ol compounds were identified at 280 nm, including epigallocatechin (EGC), catechin (C), epicatechin (EC), and epicatechin gallate (ECG). The concentrations of the identified flavane-3-ol compounds were calculated based on the calibration curve of each standard, which was achieved by injecting five different concentrations of standards separately.

### 2.10. Fluorescence Quenching Spectra Between Phenolic Compounds and BSA

Fluorescence quenching spectra between phenolic compounds and BSA were measured according to the method used in our study with a slight modification [14]. We combined 1 mL of red wine diluted with 19 mL model wine solution (12% (*v*/*v*) ethanol solution containing 3 g/L potassium hydrogen tartrate, pH3.4) as test sample, and 1 mL of test sample was added to the 9 mL 1.0 × 10^−6^ mol/L BSA buffer solution (BSA dissolved in phosphoric acid buffer solution, pH 6.8). Both excitation and emission slit widths were set at 10 nm. The samples were excited at 280 nm and the emission spectra were recorded from 285 nm to 450 nm at a scanning speed of 600 nm/min, the temperature was set at 25 °C. PBS solution (pH 6.8) was used as blank control and scanned by a Cary Eclipse fluorescence spectrometer (Agilent, Santa Clara, CA, USA).

### 2.11. Sensory Evaluation

The sensory evaluation focused on visual and taste profiles. Merlot wines were evaluated with 10 trained assessors (5 males and 5 females between the ages of 25 to 40 years old) at the College of Food and Bioengineering of Henan University of Science and Technology. Participants were required to sign a consent form to understand the motivations of sensory assessors, whether they have relevant allergy history, intake of medications impairing sensory function, as well as wearing dentures, etc. Wine samples were randomly coded and randomly assigned. The wine samples were evaluated in individual tasting rooms, using water and unsalted crackers to clean the palate. Sample wines were prepared in triplicate and in a random order at room temperature (20 ± 2 °C). Eight sensory profiles (quality, alcohol, color, clarity, bitterness, astringency, persistence, and mouthfeel) were evaluated using an unstructured linear 10 cm scale (where 0 represented “no sensation” and 10 represented “extremely high sensation”). All the evaluations were performed after 120 d aged wines, and criteria for the sensory evaluation of wine was listed in Appendix A.

### 2.12. Statistical Analysis

Data were presented as mean ± standard deviation (SD) of three replicates. Origin 2018 was used for chart drawing and SPSS 23.0 was used for statistical analysis, SIMCA 14.1 was used to perform a principal component analysis (PCA).

## 3. Results and Discussion

### 3.1. Visible Spectrum of Wine Without Pomace and Prolonged Pomace Contact Under Microwave Irradiation and Storage Time

The visible spectra of all wine samples (C, C-MW1, C-MW3, C-MW9, P, P-MW1, P-MW3, P-MW9) stored at 1 d, 20 d, 40 d, 60 d, 80 d, 100 d, and 120 d were analyzed. In Figure 1a,c, the curve profiles of C-MW1, C-MW3, C-MW9, P-MW1, P-MW3, and P-MW9 were very similar to those of the C and P samples. Moreover, under microwave irradiation, the scanning curve of wine color became higher in the visible light band. This may occur because microwave irradiation could induce the formation of free radicals in wine and further trigger chemical chain reactions [16], accelerating the formation of some stable pigments in wine after microwave treatment [23]. Furthermore, the changing trends of scanning spectra profiles of samples C, C-MW1, C-MW3, C-MW9 and P, P-MW1, P-MW3, P-MW9 were very similar, respectively. From the first day to the 120th day, the absorbance of wine without pomace increased, while the absorbance of wine with prolonged pomace contact decreased with the extension of storage time. However, the scanning curves of wine without pomace and wine with prolonged pomace contact after microwave treatment at 3 min were higher than those of wine without pomace and wine with prolonged pomace contact after microwave treatment at 1 min and 9 min, which indicated that microwave treatment at 3 min had great influence on visible spectra of wine.

Traditional aging wine was stored without pomace. The wine color turned deeper and more stable, and the scanning curve of wine color increased with the extension of aging time. The color changed during the wine’s natural aging process. On the one hand, free anthocyanins formed new oligomeric and polymeric pigments through condensation reactions. On the other hand, colorless phenols could form new color pigments through oxidative or co-pigmentation reactions, thus increasing the color intensity of wine [24]. The typical profiles of C-MW3 and P-MW3 were shown in Figure 1b,d. As shown in Figure 1b, microwave technology enhanced the profile of wine with the extension of aging time, which was similar to the change in the visible spectrum of the natural aging wine. This phenomenon suggested that microwave technology could trigger complex chemical reactions and accelerate the formation of pigments during wine storage. In other words, microwave technology could speed up the modification of wine color and shorten traditional aging time. Prolonged pomace contact was an interesting strategy for modifying the wine’s chemical composition [8]. As shown in Figure 1d, the intensity of the scanning curve of wine with prolonged pomace contact gradually decreased with the extension of aging time. This might be due to the fact that pomace, as a fining agent, adsorbed phenolic compounds or polymers [25] and reduced the contents of anthocyanins or some phenolic compounds in wine, thereby reducing the absorption intensity of the visible spectrum [8]. However, microwave treatment enhanced the profile of wine compared with the P sample, which suggested that microwave technology could accelerate the formation of stable pigments. In summary, wine samples under microwave irradiation had higher color intensity, and wine color was modified to meet the consumer’s sensor demands. Further, microwave treatment for 3 min (continuously for 15 d) is a reasonable experimental parameter for Merlot wine. The combination of microwave technology and the prolonged pomace contact strategy not only enhanced the intensity of visible spectra but also reduced the addition of traditional fining agents.

### 3.2. Changes in Color Characteristics During Storage

The browning index (BI, A_420_) represented the browning degree, which was related to the oxidation and polymerization reactions of phenolic compounds caused by non-enzymatic oxidation during wine aging. In Table 1, the wine without pomace turned into an intense yellow color with the extension of storage time. The BI of samples C, C-MW1, C-MW3, and C-MW9 increased gradually, which was similar to the changing trend in the BI during the natural aging process [26]. Furthermore, the BI of samples C-MW1, C-MW3, and C-MW9 were higher than that of sample C, which was mainly due to more free radicals induced under microwave irradiation. This lead to more oxidation and polymerization reactions of phenolic compounds caused by free radicals [16]. In addition, the BI of sample C-MW3 was higher than that of samples C-MW1 and C-MW9, which might be due to the possible degradation reactions of polymerized phenolic compounds caused by excessive free radicals [20]. This caused the BI in C-MW9 to be lower than that of C-MW3. The wine with prolonged pomace contact turned light yellow with the extension of storage time, which was due to the pomace acting as a fining agent for precipitating polymeric compounds during the aging process. So, the BI in P, P-MW1, P-MW3, P-MW9 were gradually decreased. Furthermore, the BI in P-MW1, P-MW3, and P-MW9 were higher than that of sample P, which might be due to microwave irradiation inducing more free radicals. More polymers of phenolic compounds were formed after microwave treatment [16], causing the BI under microwave irradiation to be higher than that of wine without microwave treatment. When comparing wine without pomace and wine with prolonged pomace contact, the BI in wine without pomace was higher than that of wine with prolonged pomace contact, which showed that pomace as a fining agent could decrease the BI. In contrast to wine with prolonged pomace contact, a combination of microwave technology and prolonged pomace contact strategy could enhance the BI, especially in wine after microwave treatment at 3 min.

The wine color (WC, A_520_) and color intensity (CI, the sum of A_420_, A_520_, and A_620_) of microwave-treated wines were significantly higher compared with untreated wine C and P, respectively, which indicated that microwave irradiation could significantly increase the formation of pigments, resulting in the increasing visible spectrum of wines at 420 nm, 520 nm, and 620 nm. This may be due to more free radicals being induced and chain reactions that were mediated under microwave irradiation, thereby forming more color pigments. The WC and CI of the samples C, C-MW1, C-MW3, and C-MW9 gradually increased during the storage, which owed to the gradual accumulation of color pigments and promoted the increasing visible spectrum at 420 nm, 520 nm, and 620 nm, which was similar to the changing trend of WC and CI during the natural aging process [26]. While the WC and CI in P, P-MW1, P-MW3, and P-MW9 decreased during the storage, which might due to precipitating polymeric compounds by pomace as fining agent, as well as the absorption effect of pigments by pomace during wine aging process [25]. The combination of microwave technology and prolonged pomace contact strategy not only decreased WC and CI because of the fining effect of pomace but also accelerated the changes in WC and CI in aged wine. In addition, the WC and CI of samples C-MW3 and P-MW3 were higher than those of samples C-MW1, C-MW9, and P-MW1, P-MW9, respectively. This might be due to excessive free radicals leading to the possible degradation of oxidized or polymerized phenolic compounds [16,20], causing the WC and CI of samples C-MW9 and P-MW9 to be lower than those of samples C-MW3 and P-MW3, respectively.

The *L** value of wine could indicate color intensity, and the higher *L** value, the lighter the color of the wine [11]. The *L** value in C, C-MW1, C-MW3, and C-MW9 showed a slight trend change of decreasing first and then increasing, which might be due to the polymerization reaction of phenolic compounds in the early stages of the aging process to form stable pigment compounds, causing a dark change in color intensity. So, the *L** value firstly decreased. Then, the *L** value of the wine gradually increased with the extension of aging storage, showing gradual change in color intensity from dark to light, which was similar to the increasing trend in the *L** value with the natural aging process [11]. In addition, the *L** value of wine after microwave treatment was slightly higher than that of wine without microwave treatment, which indicated microwave aging technology could improve wine clarity during wine aging process. The *L** value in P, P-MW1, P-MW3, and P-MW9 showed an obviously upward trend during wine storage, which was due to using pomace as a fining agent for clarified wine. The trend change was consistent with the wine with commercial fining agents [8]. Compared with P sample, microwave irradiation as aging technology could increase the *L** value of P-MW1, P-MW3, and P-MW9 more quickly than the C sample. Pomace could significantly increase the lightness of wine by replacing chemical fining agents or plant protein agents.

*a** represented the color tonality, and the *a** of all wine samples decreased during wine aging, which was consistent with changing trend of *a** during the natural aging process [11]. Compared with samples C and P, *a** of wines after microwave treatment decreased significantly. This demonstrated that microwave treatment could change color tonality due to microwave technology accelerating the wine aging process. *a** in wine with prolonged pomace contact was significantly decreased compared to that of wine without pomace, because pomace could effectively precipitate formed polymers during the aging process. Compared with C, C-MW1, C-MW3, and C-MW9, a combination of microwave technology and prolonged pomace contact strategy could significantly decrease the *a** of wine. From the perspective of microwave energy consumption and color tonality*,* microwave treatment for 3 min (continuously for 15 d) at aging stage was wise choice to accelerate the wine aging process.

*b** represented yellowness, and *b** increased in natural aging process of wine [11]. The *b** of all wine samples gradually increased during wine aging. This phenomenon showed that more yellow pigments formed with the extension of aging time. In addition, the *b** in C-MW1, C-MW3, C-MW9 and P-MW1, P-MW3, P-MW9 was higher than in C and P, respectively. This might be because microwave technology could enhance wine’s yellowness by inducing more free radicals [16,23]. The longer the microwave processing time, the higher the levels of *b**. *b** in wine with prolonged pomace contact was significantly increased compared to that of wine without pomace. On the one hand, this may be because phenolic compounds and flavane compounds were extracted from pomace. On the other hand, microwave technology could accelerate the formation of yellow pigments.

The *C*ab* value represented vividness, and was related to *a** and *b** [27]. The *C*ab* value gradually increased in C, C-MW1, C-MW3, and C-MW9 during wine storage, and the results indicated the color of wine without pomace tended to become vivid. In addition, there were no significant differences in *C*ab* value after microwave treatment due to the increase in *a** and the decrease in *b**, thus indicating that microwave treatment could not change the *C*ab* value of wine. However, the *C*ab* value gradually decreased in P, P-MW1, P-MW3, P-MW9 during wine storage, which might be due to pomace as a fining agent for precipitating polymeric compounds and the adsorption effect of pomace on pigments during the aging process [8,25]. Moreover, the increase in the *C*ab* value after microwave treatment at 1 min and 9 min (continuously for 15 d) indicated that wine with prolonged pomace contact tended to become vivid after microwave treatment. The reason for the increase in the *C*ab* value might be that microwave treatment facilitated the polymerization reactions of pigments derived from anthocyanins or colorless flavane compounds [11].

The behavior of *hab* tended to increase in all samples during storage, indicating that the color of all wine samples changed from red hues to brick red hues. Wine hues between 26.71° and 56.16° indicated that all wine samples were no longer fresh wine [28]. Compared with C and P, the wine hues increased after microwave treatment, and this was related to the reduction in *a** and the increase in *b**. In other words, the degradation of anthocyanins was enhanced, and yellow pigments derived from phenolic compounds were promoted under microwave irradiation [23], resulting in increasing hues. In addition, the hues of wine without pomace were lower than that of the wine with prolonged pomace contact. This might be due to the presence of pomace during the aging process. The results indicated that pomace was beneficial for decreasing pigments through precipitation, while using pomace as raw material could extract more phenolic compounds and bridged substance to form yellow pigments.

Δ*E*ab* represented the visible color difference. The results of Table 1 suggested that microwave treatment could influence the wine’s vision characteristics, and the longer microwave processing time, the greater color difference. The smaller visible color differences were caused in wine without pomace, and the bigger visible color differences were caused in wine with prolonged pomace contact, showing that the color difference in wine with prolonged pomace contact after microwave treatment was distinguishable with the naked eye, as Δ*E*ab >* 3 [11]. The results showed that pomace as a fining agent could significantly increase the visible color difference. Microwave treatment as an artificial aging technology also increased the visible color difference in wine, making wine aged further.

Overall, the color characteristics had greatly changed after 120 d of wine aging: the BI, *L**, *b**, and *hab* values gradually increased during storage, and the WC and *a** values gradually decreased during storage. These changes were consistent with common observations of visual color variations in wine. The values of the BI, WC, CI, *L**, *b**, *C***ab*, and *hab* increased after microwave treatment, while the value of *a** decreased after microwave treatment. These changes indicated that microwave technology had the effect of enhancing wine color characteristics and accelerating the wine aging process. The values of the BI, WC, CI, *a**, and *C***ab* in wine with prolonged pomace contact were lower than those of wine without pomace. These changes showed that color characteristics in wine with prolonged pomace contact tended to flatten. The values of the *L**, *b**, *hab*, and Δ*E*ab* in wine with prolonged pomace contact were higher than those of the wine without pomace. These changes showed that the presence of pomace in wine could make wine appear yellow and light with the naked eye. From the perspective of microwave energy consumption and effects of color characteristics*,* microwave treatment for 3 min (continuously for 15 d) was a wise choice at the wine aging stage.

### 3.3. Changes in TPC, TMA, TT, and TF_3-ols_ During Storage

As shown in Figure 2, the TPC, TMA, TT, and TF_3-ols_ of all wine samples showed an overall downward trend during storage, which was similar to the changing trend in natural aging wine [26]. In Figure 2a,b, the TPC in all samples had decreased overall, although it fluctuated in wine with prolonged pomace contact at 40 d. The decrease in TPC might be due to the formation of dimers or polymers between phenolic compounds and electrophilic quinones, which were further rearranged and oxidized to further polymerization, resulting in a decrease in TPC in wine during storage [26]. The slight increase in TPC in wine with prolonged pomace contact might be due to the complete extraction of phenol from the skin and seeds on the 40th day of pomace contact. Furthermore, the TPC in microwave treated wines was higher than that of C and P. On the one hand, the increase in TPC might be due to the breaking of hydrogen bonds in dimers or polymers caused by more free radicals induced by microwave irradiation [16]. On the other hand, the increase in TPC might be due to the inhibition of polyphenol oxidase activity under microwave irradiation, leading to the reduction in brown oxidation of phenolic compounds [14]. The TPC in wine with prolonged pomace contact was lower than that of wine without pomace, which was because the adsorption of phenolic compounds by the pomace exceeded extraction of phenolic compounds from pomace. So, prolonged pomace contact during the aging stage was not beneficial for increasing the TPC. The TPC of C-MW1 and P-MW1 were higher than those of C-MW3, C-MW9 and P-MW3, P-MW9, respectively. This phenomenon indicated that the generation of more free radicals induced by longer microwave treatment might lead to the degradation and structural damage of polymerized polyphenols. So, short-time microwave treatment could increase the TPC in Merlot wine. The TPC in C-MW1, C-MW3, C-MW9 were higher than in that of P, which showed that the combination of microwave technology and prolonged pomace contact strategy could significantly decrease the contents of TPC, especially through microwave treatment for 9 min (continuously for 15 d).

TMA, which played an important role in wine color stability, polymerized or reacted with other compounds to form more stable compounds [29]. So, the contents of TMA decreased during the wine aging process, and a gradual decrease in TMA in all samples was also observed during storage, as shown in Figure 2c,d. The concentrations of TMA in microwave-treated samples were lower than in C and P, respectively, showing that microwave technology could accelerate the polymerization and reaction of anthocyanins with other compounds and form more stable pigments. In other words, microwave technology accelerated the wine aging process. The longer the microwave-treated time, the lower the contents of TMA. The reason for microwave-accelerated TMA polymerization might be related to the formation of more 1-hydroxyethyl free radicals after longer time microwave treatment [16], especially microwave treatment for 9 min (continuously for 15 d). The contents of TMA in P, P-MW1, P-MW3, and P-MW9 were lower than in that of C. This might be due to the combination of microwave technology and prolonged pomace contact strategy. Not only does the aging effect of microwave technology reduce the contents of TMA, but also increaes the precipitation and absorption of TMA or polymeric anthocyanins by pomace [8,25].

The TT increased the friction of the tongue and mouth surfaces, creating a dry or rough feeling, and contributing to astringency in wine. Condensed tannins in wine were derived from seed and skin, and hydrolyzable tannins were derived from wood cooperage. The reason for the decrease in tannins contents was that tannins formed insoluble compounds with cell walls’ polysaccharides or structural proteins [30]. As shown in Figure 2e,f, the contents of TT in all wine samples gradually decreased during the wine aging process. The concentrations of TT in microwave-treated samples were lower than those of C and P, respectively. The reason might be because microwave technology accelerated the formation of tannins with cell walls polysaccharides or structural proteins. The contents of TT in samples C, C-MW1, C-MW3, C-MW9 were higher than that of P. The reasons for this might be that the effect of pomace as a fining agent on precipitation tannins exceeded the effect of microwave extraction of tannins from seed and skin. Microwave aging technology could accelerate the decrease in the content of tannins, which could reduce the astringency and improve the wine mouthfeel. The TT of C-MW3 and P-MW3 were lower than those of C-MW1, C-MW9 and P-MW1, P-MW9, respectively. This phenomenon indicated that microwave treatment at 3 min (continuously for 15 d) could better accelerate the wine aging process.

TF_3-ols_ were commonly found in wine, and were responsible for wine’s astringency and bitterness. Figure 2g,h shows a decreasing trend in concentrations of TF_3-ols_ during the aging process, and similar results were observed in Merlot wine [31]. The reduction of TF_3-ols_ was probably triggered by the oxidation and polymerization reactions to form oligomers or polymers during the wine aging process. In addition, the concentrations of TF_3-ols_ in wine with prolonged pomace contact were significantly lower than those of wine without pomace, which might be related to the greater formation of TF_3-ols_ aggregates or the adsorption of pomace as a fining agent. The concentrations of TF_3-ols_ in wine with prolonged pomace contact were significantly lower than 600–1000 mg/L in commercial wine [32]. This indicated that prolonged pomace contact had no positive effect on bitterness and astringency in wine. As shown in Figure 2g,h, the concentrations of TF_3-ols_ in wine after microwave treatment at 3 min (continuously for 15 d) rapidly decreased compared with other samples. This showed that the oxidation and polymerization reactions of TF_3-ols_ were significantly accelerated after microwave treatment at 3 min.

### 3.4. Identification, Quantification, and Analysis of Flavane-3-ol Monomers During Storage

Epigallocatechin (EGC), catechin (C), epicatechin (EC), and epicatechin gallate (ECG) were the main flavan-3-ols monomers associated with bitterness and astringency in wine [32]. Using the above HPLC method for flavane-3-ol monomers analysis in Figure 3, the contents of flavane-3-ol monomers in all wine samples were determined, including EGC, C, EC, and ECG. As shown in Figure 3a,b, the contents of flavanol monomers (C, EC, and ECG) decreased during the aging period, while the content of EGC showed a fluctuating trend during the aging period. The content of C was the highest, EGC was the second highest monomer, followed by the content of EC. The contents of EGC was the lowest in all wine samples. The intensity of astringency and bitterness were influenced by flavane-3-ol monomers, not only in their concentration, but also in their structure [33]. Petropoulos et al. [34] reported that bitterness was positively correlated with EC concentration and negatively correlated with EGC concentration. Flavane-3-ol monomers tended to form flavanol dimers by either C4-C8 or C4-C6 inter-flavanols bond [33], or flavanol and anthocyanin via ethyl bridge to form ethyl-linked anthocyanin-flavanol pigments; so, the concentration of flavanol oligomers/polymers increased in aged wines [35]. Many researchers have reported that the bitterness of wine generally comes from flavane-3-ol monomers, while the astringency of wine was positively correlated with the degree of flavane-3-ol polymerization [33]. For EGC, C, and EC, all the treated wines exhibited lower contents than control wine during storage. This might be because microwave treatment accelerated the polymerization reactions of EGC, C, and EC. Our research team also found that under microwave irradiation, the formation of catechin polymers was accelerated due to the induction of more free radicals in the model wine solution [16]. For ECG, all the treated wines exhibited slightly higher content than control wines at the same storage stage. This may be because microwave treatment accelerated the formation of ECG from epicatechin and gallic acid in wine. For C, EC, and ECG, the concentrations in wine with prolonged pomace contact exhibited lower contents than those of wine without pomace. This might be because the adsorption effect of pomace as a fining agent was better than the extraction effect on C, EC, and ECG from skins and seeds, while the result of EGC was the opposite effect. The effect of microwave treatment at 3 min was significantly better than that of microwave treatment at 1 min or 9 min in Figure 3a,b. In other words, microwave technology could accelerate wine aging by rapidly changing the concentrations and structures of flavane-3-ol monomers, thereby quickly reducing the wine’s bitterness and enhancing its astringency. The combination of microwave technology and the prolonged pomace contact strategy could clearly decrease wine’s bitterness; however, this requires further research.

### 3.5. Fluorescence Quenching of Wine Samples and BSA

The astringency derived from the precipitation was formed by a hydrophobic interaction between salivary proteins and astringent substances via hydrogen bonding in the oral cavity. The contents and structures of astringent substances (including tannins and polyphenols) could affect the wine astringency intensity [36], and polysaccharides also could modify astringency via the formation of a ternary mixture with proteins-flavan-3-ols through hydrophobic interactions [32]. To better understand the changes in the astringency profile of wine, BSA was employed to study the interaction with astringent substances. Fluorescence quenching was the decrease in the fluorescence quantum yield caused by an interaction between fluorophore and quencher molecules. In other words, the results of fluorescence quenching were caused by an interaction between BSA and astringent substances or polysaccharides in wine. In Figure 3c,d, fluorescence intensity gradually increased with the extension of wine aging, indicating that the interaction between BSA and astringent substances gradually decreased. In other words, the contents of tannins and polyphenols perhaps decreased or the structures had changed. The result of decreasing contents of TT was consistent with the report in Figure 2e,f. After different microwave treatment times, the fluorescence intensity of wine with prolonged pomace contact was slightly higher than that of wine without pomace. This phenomenon might be due to the absorption effects of pomace on tannins, phenolic compounds, or polysaccharides being slightly higher than the extraction effect of tannins and polyphenols from seed and skin. As shown in Figure 3e,f, the fluorescence intensity after microwave treatment at 9 min was lowest, indicating that the astringency profile was strongest at 9 min (continuously for 15 d). The combination of microwave technology and the prolonged pomace contact strategy had an influence on fluorescence quenching. On the one hand, prolonged microwave treatment could extract more tannins and polyphenols from seed and skin. On the other hand, some tannins and polyphenols might be desorbed from pomace due to mechanical vibration caused by microwave, so the combined technology could slightly decrease the astringent of wine compared to single microwave aging technology.

### 3.6. Principal Component Analysis and Sensory Evaluation

In order to investigate the effects of microwave technology and prolonged pomace contact technology on Merlot wine, principal component analysis (PCA) was used to analyze the correlation load maps of color and mouthfeel characteristics in samples (C, C-MW1, C-MW3, C-MW9, P, P-MW1, P-MW3, and P-MW9) from 1 to 120 days in Figure 4a. Here, 80.8% variance was explained by 17 color and mouthfeel parameters. The biplot of PC1 and PC2 accounted for 73.2% and 7.6% of the data variables. Wine with prolonged pomace contact and wine without pomace were clustered separately, and the distribution of wine samples implied that wine both with and without pomace had a positive impact on the sensory characteristics of Merlot wine. Wines without pomace contact were located at the upper and lower right quadrant, while wines with pomace contact were located at the upper and lower left quadrant. C wine was clustered tightly for TT and TMA, which indicated that the astringency and color of naturally aged wine was evident; while BI, WC, CI, *a**, *C*ab*, TPC, TF_3-ol_, C, EC, and ECG were tightly clustered in microwave treatment wines, which illustrated that microwave treatment could accelerate the color change in Merlot wine, reduce bitterness, and enhance softness. In other words, microwave technology could accelerate the wine aging process. P wine was clustered for EGC, which indicated that adding pomace wine rendered it rich in the astringent compound EGC; while *L**, *b**, Δ*E*ab*, *hab*, and EGC were tightly clustered in with pomace contact wine after microwave treatment, which indicated microwave technology could help to the lightness, yellowness, color difference, and bitterness of adding pomace wine. From the PCA shown in Figure 4a, it can be seen that adding pomace during wine aging had a positive effect on the clarity of the wine, and microwave technology could quickly change color and mouthfeel characteristics during wine aging.

Figure 4b showed the sensory evaluation of eight wine samples by 10 trained assessors. The scores of sensory profiles ranged from 2.8 to 8.7 points. There are differences in the degree of sensory profiles in different treatments, especially between wine without pomace and wine with prolonged pomace contact. Microwave treatment could improve wine quality, color, clarity, astringency, persistence, and mouthfeel profiles, especially combined with 3 min (continuously for 15 d) microwave treatment, while microwave treatment decreased the wine bitterness profile. In terms of alcohol, the performance of microwave-treated wine was basically unchanged, which might be because low-temperature microwave treatment did not significantly change the content of alcohol. Microwave treatment combined with prolonged pomace contact improved the wine clarity profile and offset part of the effect of prolonged pomace contact on the wine’s color, persistence, and mouthfeel profiles. In summary, microwave treatment had a positive effect on wine sensory profiles, and the combination of microwave technology and the prolonged pomace contact strategy also had a great influence on the sensory profiles.

## 4. Conclusions

The combination of microwave irradiation technology and prolonged pomace contact changed the color characteristics and mouthfeel characteristics of Merlot wine. The current study provided some information about prolonged pomace contact. The results showed that the clarification and yellowness of wine increased, but the bitterness and astringency reduced due to the decrease in TPC, TT, TF_3-ol,_ related to bitterness and astringency. The contents of pigments and TMA decreased due to the fining effect of pomace, resulting in the loss of color intensity. Therefore, the technique of prolonged pomace contact should not only consider the beneficial effects, but also consider the effects on color and mouthfeel. However, microwave irradiation had a long-term impact on the evolution of wine color and mouthfeel characteristics. Related detected parameters also exhibited a similar trend change to the natural aging process. The results indicated that appropriate microwave-treated time and frequency could accelerate some aging reactions related to color and mouthfeel characteristics, which proceed slowly in untreated wine. In particular, microwave treatment for 3 min or 9 min (continuously for 15 d) had a significant influence on wine color and mouthfeel characteristics. In summary, the fining and extraction effects of pomace could not offset the side effects on color and mouthfeel characteristics. But, the potential positive effects on color and mouthfeel in Merlot wine due to the application of microwave irradiation had been confirmed. The changes in other wines of different varieties under microwave irradiation and prolonged pomace contact have not yet been studied. The changes in color and mouthfeel characteristics in Merlot wine remain worth studying, and changing wine characteristics should be further investigated on higher quality wine using microwave technology.

## Data Availability

The original contributions presented in the study are included in the article/Appendix A, further inquiries can be directed to the corresponding author.

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
