# Peer review of "Effects of Prolonged Pomace Contact on Color and Mouthfeel Characteristics in Merlot Wine During the Ageing Process Under Microwave Irradiation"

_foods, 2025, doi:10.3390/foods14030507_

Round 1

Reviewer 1 Report

Comments and Suggestions for Authors

Dear authors,

I analyzed the manuscript and these are the observations that should be taken into account.

1.     The topic addressed is a well-studied one in the field of oenology. So, it cannot be clearly observed what would be the innovation and relevance of the analysis made by the authors. Otherwise, the bibliographic list is not enough and should be deeply analyzed, especially review-type works:

https://www.mdpi.com/2076-3417/14/17/7537#B61-applsci-14-07537

https://www.mdpi.com/2304-8158/11/12/1778

https://www.bio-conferences.org/articles/bioconf/full_html/2017/02/bioconf-oiv2017_02032/bioconf-oiv2017_02032.html

2.     The abstract must be redone and the following must be taken into account:

Ø  include specific changes observed in colour and mouthfeel parameters and key metrics from sensory evaluations,

Ø  briefly summarize the principal components that were significant,

Ø  address typographical errors and restructure some sentences for clarity and conciseness,

Ø  compare microwave treatment to traditional aging methods,

Ø  emphasizing time savings, resource efficiency, or sensory quality improvements.

3.     All the methods used must be described, naming the laboratory equipment used and the way of working, even if in a succinct way.

4.     The work must be proofread by an English language professional to give clarity to the text.

5.     In order to give a relevant scientific importance to the subject, it is good to consider the following:

Ø  While the effects of microwave irradiation are mentioned, the mechanism of non-thermal effects remains vague and underexplained, weakening scientific rigor.

Ø  Insufficient detail on the interaction between pomace components and microwaves during ageing.

Ø  Include comparisons with other ageing and fining technologies to highlight advantages and limitations.

Ø  Provide specific metrics on time, cost savings, and sustainability benefits to enhance industrial relevance.

Ø  Narrow the scope to provide deeper insights into the synergy between microwave irradiation and pomace contact.

Ø  Compare microwave treatment to traditional aging methods, emphasizing time savings, resource efficiency.

Comments on the Quality of English Language

The work must be proofread by an English language professional to give clarity to the text.

Author Response

Dear reviewer,

Thank you very much for giving us the opportunity to improve our manuscript (Manuscript: Foods-3416176). Your suggestions and comments were all valuable and very helpful for revising and improving our paper. We have given careful consideration to each point that was raised. We have restructured the abstract and conclusion, improved descriptions of the methods, provided deeper insights on combination of microwave irradiation and prolonged pomace contact strategy, provided specific metrics of microwave ageing technology, and revised grammar. In this revised version, changes to our manuscript within the document were all highlighted by using red colored text. Below, we have provided our response to each comment.

Reviewer 2 Report

Comments and Suggestions for Authors

Dear authors,

Your research on the effects of prolonged maceration and microwave irradiation treatment on the physicochemical properties of wine makes a valuable contribution to the advancement of functional food studies. The work is written concisely and is easy to follow, and the presentation of the figures is very clear.

However, I have a few suggestions that could further enhance the quality of the presentation.

Firstly, although the descriptions of the methods in sections 2.5, 2.6, 2.7, and 2.8 reference the cited sources, it would improve clarity if the procedures for each method were briefly outlined.

Additionally, Table 1 could be made clearer by placing the letters indicating significance next to the numbers instead of below them.

Author Response

Dear reviewer,

Thank you very much for giving us the opportunity to improve our manuscript (Manuscript: Foods-3416176). Your suggestions and comments were all valuable and very helpful for revising and improving our paper. We have given careful consideration to each point that was raised. We have improved descriptions of the methods and changed Table 1. In this revised version, changes to our manuscript within the document were all highlighted by using red colored text. Below, we have provided our response to each comment.

Respond to Reviewer #2’ comments

  1. Firstly, although the descriptions of the methods in sections 2.5, 2.6, 2.7, and 2.8 reference the cited sources, it would improve clarity if the procedures for each method were briefly outlined.

Response: According to the reviewer’s suggestion, we have supplemented the descriptions of experimental methods in sections 2.5, 2.6, 2.7, and 2.8, please see line 173-176, 180-189, 192-196, 199-204, respectively.

And we have supplemented descriptions of laboratory equipment in sections 2.2 and the descriptions of experimental methods in sections 2.4, please see line 141-146, 170, respectively.

  1. Additionally, Table 1 could be made clearer by placing the letters indicating significance next to the numbers instead of below them.

Response: According to the reviewer's suggestion, the letters have been revised in Table 1.

Round 2

Reviewer 1 Report

Comments and Suggestions for Authors

Dear authors. 

You have largely made the required corrections. It would be interesting to add more bibliographical titles, taking into account that there have been multiple scientific studies on this topic.